# Effect of Transarterial Chemotherapy on the Structure and Function of Gut Microbiota in New Zealand White Rabbits

**DOI:** 10.3390/biology13040230

**Published:** 2024-03-29

**Authors:** Liuhui Bai, Xiangdong Yan, Ping Qi, Jin Lv, Xiaojing Song, Lei Zhang

**Affiliations:** 1The First Clinical Medical College, Lanzhou University, Lanzhou 730030, China; bailh21@lzu.edu.cn (L.B.); yanxd21@lzu.edu.cn (X.Y.); qip21@lzu.edu.cn (P.Q.); lvj21@lzu.edu.cn (J.L.); songxiaojing4227@126.com (X.S.); 2Department of General Surgery, The First Hospital of Lanzhou University, Lanzhou 730030, China; 3Key Laboratory of Biotherapy and Regenerative Medicine of Gansu Province, The First Hospital of Lanzhou University, Lanzhou 730030, China

**Keywords:** chemotherapy, gut microbiota, hepatocellular carcinoma, transcatheter arterial chemoembolization, intestinal permeability, adverse reaction

## Abstract

**Simple Summary:**

Gut microbiota (GM) are closely related to hepatocellular carcinoma (HCC) occurrence and development. Furthermore, patients with HCC who have received transcatheter arterial chemoembolization (TACE) treatment often experience adverse gastrointestinal reactions, which may be related to changes in the GM caused by the chemotherapeutic drugs used in TACE. We analyzed changes in the GM of New Zealand white rabbits treated with hepatic arterial chemotherapy by measuring the levels of serological and colonic tissue markers. Simultaneously, we evaluated the correlation between the GM and these markers to explore the mechanism by which chemotherapy affects the GM. Following transarterial chemotherapy with epirubicin, the Firmicutes abundance decreased, whereas that of Proteobacteria increased, and beneficial bacteria decreased, while harmful bacteria increased. Following chemotherapy, the GM of rabbits showed a dynamic change over time, first aggravating and then subsiding. The changes were most notable on the fourth day after surgery and recovered slightly on the seventh day.

**Abstract:**

The gut microbiota (GM) are closely related to hepatocellular carcinoma (HCC) occurrence and development. Furthermore, patients with HCC who have received transcatheter arterial chemoembolization (TACE) treatment often experience adverse gastrointestinal reactions, which may be related to changes in the GM caused by the chemotherapeutic drugs used in TACE. Therefore, we conducted animal experiments to investigate these changes. We analyzed changes in the GM of New Zealand white rabbits treated with hepatic arterial chemotherapy by measuring the levels of serological and colonic tissue markers. Simultaneously, we evaluated the correlation between the GM and these markers to explore the mechanism by which chemotherapy affects the GM. Following transarterial chemotherapy with epirubicin, the Firmicutes abundance decreased, whereas that of Proteobacteria increased. The relative abundance of beneficial bacteria, such as *Muribaculaceae*, *Enterococcus*, *Ruminococcus*, and *Clostridia*, decreased in the experimental group compared with those in the control group. However, the relative abundance of harmful bacteria, such as *Bacteroides* and *Escherichia* (*Shigella*), was higher in the experimental group than in the control group. Following chemotherapy, the GM of rabbits showed a dynamic change over time, first aggravating and then subsiding. The changes were most notable on the fourth day after surgery and recovered slightly on the seventh day. The changes in the host’s GM before and after arterial chemotherapy are evident. Hepatic arterial chemotherapy induces dysbiosis of the intestinal microbiota, disrupts intestinal barrier function, damages the integrity of the intestinal mucosa, increases intestinal permeability, facilitates excessive passage of harmful substances through the gut–liver axis communication between the liver and intestine, and triggers activation of inflammatory pathways such as LPS-TLR-4-pSTAT3, ultimately leading to an inflammatory response. This study provides a theoretical basis for combining TACE with targeted GM intervention to treat HCC and reduce adverse gastrointestinal reactions.

## 1. Introduction

Primary liver cancer presents high morbidity and mortality rates in China, threatening the lives and health of many people. Primary liver cancer includes three different pathological types: hepatocellular carcinoma (HCC), intrahepatic cholangiocarcinoma (ICC), and mixed type (HCC-ICC). Among them, HCC accounts for 85–90% of all primary liver cancer cases [1]. Owing to its hidden onset and lack of specific symptoms in the early stages, most patients with HCC in China are diagnosed in the middle and advanced stages and miss the opportunity for surgical curative treatment. Chemotherapies, such as transcatheter arterial chemoembolization (TACE) and hepatic arterial infusion chemotherapy, play important roles in HCC treatment; both of these use a high concentration of chemotherapeutic drugs locally administered to kill cancer cells, which also damages some normal liver cells [2]. Of the TACE treatments, epirubicin, oxaliplatin, and fluorouracil are the most commonly used, either alone or in combination. Although the actual efficacy of TACE is relatively clear, it easily causes adverse gastrointestinal reactions, such as post-operative nausea and vomiting [3,4].

The gut microbiota (GM) play important roles in the occurrence, development, diagnosis, and treatment of tumors [5]. Abuse of antibiotics, surgery, chemotherapy, environmental changes, and other factors can cause GM dysbiosis by reducing host immunity or directly damaging the intestine. Dysbiosis refers to abnormal changes in the type, quantity, and proportion of the GM that deviate from a normal physiological combination, which develop into a pathological state [6]. GM dysbiosis leads to an imbalance in the intestinal microecosystem, characterized by a decrease in beneficial bacteria and an increase in harmful bacteria. GM dysbiosis manifests clinically as nausea, abdominal pain, diarrhea, and other symptoms [6].

The diversity of the GM in patients with HCC after TACE treatment are reduced, evidenced by a decrease in the relative abundance of probiotics, such as Firmicutes and Bacteroidetes, and an increase in the relative abundance of pathogenic bacteria, such as Proteobacteria, Fusobacteria, and opportunistic bacteria [7]. Simultaneously, these patients are prone to nausea, abdominal pain, diarrhea, and other gastrointestinal reactions, which may be related to an imbalance in microbiota caused by the chemotherapy drugs used in TACE treatment [8]. In treating HCC with TACE, epirubicin, a specific anticancer drug, has similar clinical efficacy, gastrointestinal reactions, and overall survival rate to commonly used chemotherapeutic drugs, such as oxaliplatin and fluorouracil; however, the recurrence rate of epirubicin is low, tolerance is good, and drug resistance is relatively low [9,10].

Based on the above findings, we administered an epirubicin suspension to New Zealand white rabbits via the hepatic artery. Changes in the GM before and after chemotherapy were analyzed and compared, and the correlations among these changes, the related For this section please double check that the following details are provided for all cases: company names and addresses (city, country) of the instruments and agents, version numbers of the softwares used. For USA or Canadian companies, please provide the state name as well (i.e., city name, abbreviated state name, USA/Canada). Same for the following highlights, please confirm. inflammatory factors, and intestinal barrier proteins were explored. This study aimed to provide novel treatment insights for clinical patients with HCC and reduce adverse reactions following TACE.

## 2. Materials and Methods

### 2.1. Equipment

Sodium pentobarbital solution (3%), epirubicin suspension (10 mg in 1 mL normal saline), and raw materials were purchased from the First Hospital of Lanzhou University (Lanzhou, China). Tissue scissors, hemostatic forceps, forceps, arterial clips, suture needles, syringes, gauze, and other surgical instruments were provided by the laboratory of the First Hospital of Lanzhou University.

### 2.2. Animals

Male New Zealand white rabbits (2.5–3.5 kg) were purchased from the Experimental Animal Center of Lanzhou University (Lanzhou, China) and bred according to the standard conditions of the National Animal Care and Utilization Committee. The rabbits were adaptively housed for 1 week.

### 2.3. Operation

Thirty New Zealand white rabbits were randomly divided into the following five groups (*n* = 6 per group): control on post-operative day 1 (D1d), control on post-operative day 7 (D7d), chemotherapy on post-operative day 1 (T1d), chemotherapy on post-operative day 4 (T4d), and chemotherapy on post-operative day 7 (T7d). One rabbit perished in both the T1d and D7d groups during the modeling procedure. Rabbits were anesthetized using a 3% pentobarbital sodium solution (30 mg/kg) injection via the auricular vein and routinely separated using towels. The proper hepatic artery was found and separated by laparotomy, and a 1.5 mg/kg epirubicin suspension (10 mg epirubicin in 1 mL normal saline) was injected. After observing no bleeding points, the rabbits were sutured layer-by-layer, bandaged, and fixed. In the control group, the epirubicin suspension was replaced with a similar dose of normal saline, and the rest of the procedure was the same as that in the experimental group. The rabbits were anesthetized with a 3% sodium pentobarbital solution at the corresponding time points after surgery for serum, stool, and colon tissue sample collection. After sampling, the rabbits were euthanized via air embolization through the ear vein. The samples were washed with normal saline and stored at −80 °C until further analysis. The animal experimental protocol was approved by the Ethics Committee of Lanzhou University (LDYYLL2024-11).

### 2.4. Microbial Bioinformatics Analysis

#### 2.4.1. Stool Sample Processing and DNA Extraction

The fecal samples collected were frozen immediately in liquid nitrogen and stored at −80 °C until analysis. Fecal genomic DNA was extracted from the fecal samples using the QIAamp^®^ DNA Stool Mini Kit (Qiagen, Hilden, Germany), according to the manufacturer’s protocol. The concentration and purity were detected using a Nanodrop, and the integrity was assessed using regular 0.8% agarose gel electrophoresis.

#### 2.4.2. High-Throughput Sequencing

The bacterial genomic DNA was used as the template to amplify the V3–V4 hypervariable region of the 16S rRNA gene with the forward primer (5′-CCTACGGGNGGCWGCAG-3′) and the reverse primer (5′-GACTACHVGGGTATCTAATCC-3′). Each sample was independently amplified thrice. Finally, the polymerase chain reaction (PCR) products were assessed by agarose gel electrophoresis and the PCR products from the same sample were pooled. The pooled PCR product was used as a template, and index PCR was performed using index primers to add the Illumina index to the library. The amplification products were verified by gel electrophoresis and purified using an Agencourt AMPure XP Kit (Beckman Coulter, CA, USA). The purified products were indexed to the 16S V3–V4 library. The library quality was assessed using a Qubit@2.0 Fluorometer (Thermo Scientific, Massachusetts, MA, USA) and an Agilent Bioanalyzer 2100 system. Finally, the pooled library was sequenced on an Illumina MiSeq 250 Sequencer to generate 2 × 250 bp paired-end reads.

#### 2.4.3. Bioinformatics Analysis

The raw reads were quality-filtered and merged using the following criteria: (1) truncation of the raw reads at any site with an average quality score < 20, removal of reads contaminated by the adapter, and further removal of reads having less than 100 bp by TrimGalore; (2) the paired-end reads were merged to tags using Fast Length Adjustment of Short reads (FLASH, v1.2.11); (3) removal of reads with ambiguous bases (N base) and reads with more than 6 bp of homopolymers by Mothur; (4) removal of reads with low complexity to obtain clean reads for further bioinformatics analysis. The remaining unique reads were chimera-checked, compared with the gold.fa database (http://drive5.com/uchime/gold.fa), accessed on 18 October 2023, and clustered into operational taxonomic units (OTUs) using UPARSE with a 97% similarity cut-off. All OTUs were classified based on the Ribosomal Database Project (RDP) Release 9 201203 by Mothur. Rarefaction and alpha diversity (including the Shannon, Simpson, and InvSimpson indices) were analysed using Mothur. Using R project (Vegan software package, V3.3.1), the sample trees were clustered by the Bray–Curtis distance matrix, and the data were analyzed by using the unweighted pairing method of arithmetic average (UPGMAO), weighted unifrac, unweighted unifrac, Bray–Curtis, and Jaccard principal coordinate analysis (PCoA), based on OTUs. Redundancy analysis (RDA) was performed using Canoco for Windows 4.5 (Microcomputer Power, New York, NY, USA) and assessed using MCPP with 499 random permutations.

### 2.5. Enzyme Linked Immunosorbent Assay and D-Lactic Acid Microplate Method

Alanine aminotransferase (ALT), aspartate aminotransferase (AST), diamine oxidase (DAO), D-lactic acid (D-LA), secretory immunoglobulin A (sIgA), tumor necrosis factor-α (TNF-α), interleukin (IL)-6, IL-10, cyclooxygenase-2 (COX-2), lipopolysaccharide (LPS), Toll-like receptor 4 (TLR-4), phosphorylated signal transducer and activator of transcription 3 (pSTAT3), phosphorylated nuclear factor κb (pNF-KB), giant cell colony-stimulating factor (GM-CSF), NOD-like receptor pyrin domain-containing protein 3 (NLRP3), mucin 2 (MUC2), tight junction protein 1 (ZO-1), occludin, and claudin-1 levels were detected in serum and colon tissues and analyzed according to manufacturer’s instructions. This was performed using the following kits: rabbit ALT enzyme-linked immunosorbent assay (ELISA; ml027206, Yuanju, Shanghai, China), rabbit AST ELISA (ml036743, Yuanju, Shanghai, China), rabbit DAO ELISA (ml027888, Meilian, Shanghai, China), rabbit sIgA ELISA (ml036798, Meilian, Shanghai, China), rabbit TNF-α ELISA (ml027766, Yuanju, Shanghai, China), rabbit IL-6 ELISA kt (ml125512, Meilian, Shanghai, China), rabbit IL-10 ELISA (ml332541, Meilian, Shanghai, China), rabbit COX-2 ELISA (ml3332512, Meilian, Shanghai, China), rabbit LPS ELISA (ml44523, Meilian, Shanghai, China), rabbit TLR-4 ELISA (ml2254156, Meilian, Shanghai, China), rabbit pSTAT3 ELISA (ml1144586, Meilian, Shanghai, China), rabbit pNF-KB ELISA (ml459985, Meilian, Shanghai, China), rabbit GM-CSF ELISA (ml421125, Yuanju, Shanghai, China), rabbit NLRP3 ELISA (ml665211, Meilian, Shanghai, China), rabbit MUC2 ELISA (ml555632, Meilian, Shanghai, China), rabbit ZO-1 ELISA (ml027943, Meilian, Shanghai, China), rabbit Occludin ELISA (ml005521, Meilian, Shanghai, China), rabbit Claudin-1 ELISA (ml666985, Meilian, Shanghai, China), and D-LA content detection (ml622365, Yuanju, Shanghai, China).

### 2.6. Statistical Analyses

Statistical analysis was performed using SPSS 26.0 (IBM, Armonk, NY, USA) software. To determine the statistical differences between the two groups, we used an independent samples *t*-test and the Mann–Whitney U test. And the Spearman Rank correlation test was used for correlation analysis. The R package vegan (v2.5.6) was used for alpha diversity and species composition analysis. The R package VennDiagram (v1.6.20) was used for Venn diagram analysis; the R package ade4 (v1.7.13) was used for beta diversity NMDS analysis; the R package pheatmap (version v1.0.12) was used for environmental factor and correlation analysis. *p* < 0.05 was considered statistically significant (* *p* < 0.05, ** *p* < 0.01). Analysis of variance was used to analyze the differences in various indicators (*p* < 0.05).

## 3. Results

### 3.1. Alpha Diversity and Species Composition of the Microbiota

Alpha diversity analysis was performed based on OTUs. In this study, we assessed the intestinal microbial diversity in rabbits using different diversity indices (observed, ACE, Chao1, Shannon, Simpson, and PD). Compared with control group D1d, statistical analysis of intestinal flora Alpha diversity index in experimental group T1d showed that there was no significant difference in the richness and diversity of intestinal flora in New Zealand white rabbits after hepatic arterial chemotherapy (*p* > 0.05). Compared with the control group on day 7, the Alpha diversity index analysis of the intestinal flora in the experimental group on day 7 showed that the richness and diversity of the intestinal flora in New Zealand white rabbits after hepatic arterial chemotherapy were decreased, and the difference was statistically significant (*p* < 0.05) (Figure 1). 

R package VennDiagram (v1.6.20) used to calculate the number of OUTs in each group. A total of 3786 OUTs were identified in the D1d group, of which 629 were unique; a total of 3557 OUTs were identified in the T1d group, of which 400 were unique; a total of 3648 OUTs were identified in the D7d group, of which 789 were unique; and a total of 3395 OUTs were identified in the T7d group, of which 536 were unique. This indicated that the number of bacterial species in the experimental group was lower than that in the control group (Figure 2).

The species abundance of each sample was integrated and analyzed at different taxonomic levels and microbiota composition was visually displayed using bar charts. At the phylum level, Firmicutes was the most abundant and common phylum among all groups, followed by Bacteroidetes, Verrucomicrobia, and Proteobacteria. The Firmicutes abundance decreased in the experimental group, while that of Proteobacteria increased compared with those in the control group (Figure 3). At the genus level, the relative *Muribaculaceae* abundance was higher in the control group than that in the experimental group and this was the dominant bacterium. The relative abundance of beneficial bacteria, such as *Muribaculaceae*, *Enterococcus*, *Ruminococcus*, and *Clostridia*, decreased in the experimental group compared with those in the control group. However, the relative abundance of harmful bacteria, such as *Bacteroides* and *Escherichia* (*Shigella*), was higher in the experimental group than that in the control group (Figure 4).

### 3.2. Microbiota Beta Diversity

Nonmetric multidimensional scaling (NMDS) and permutational multivariate analysis of variance (PerMANOVA) showed a significant difference in bacterial community composition between the D1d and T1d groups (*p* = 0.007), as well as between the D7d and T7d groups (*p* = 0.008) (Figure 5). These findings suggest that chemotherapy intervention significantly altered the composition of the GM communities in rabbits.

### 3.3. Differential Bacterial Species

Difference analysis was used to screen species with significant differences between the two groups and display the significantly enriched bacteria of each group on a species taxonomic branch map according to different classification levels, such as phylum, class, order, family, and genus (Figure 6). Linear discriminant analysis (LDA) was used to assess each value of the significant difference in the effect of species (i.e., LDA score), and |LDA| > 2 and *p* < 0.05 was used as the difference screening threshold to obtain the species with significant differences in relative abundance between groups (Figure 7).

### 3.4. Serum Differences and Tissue Indices

The serum levels of ALT, AST, DAO, sIgA, TNF-α, IL-6, COX-2, LPS, pSTAT3, GM-CSF, and NLRP3 in the experimental group were significantly higher than those in the control group. The levels of IL-10, ZO-1, occludin, and claudin-1 were significantly decreased, while the levels of MUC2, pNF-KB, and TLR-4 were not significantly changed. The levels of ZO-1, occludin, and claudin-1 were also significantly decreased in the colon tissues compared with those in the control group. The serum D-LA content detected by D-LA colorimetry was significantly increased in the experimental group compared with that in the control group (Figure 8). 

### 3.5. Correlation Analysis of Bacteria and Indicators of Each Group

Based on Spearman’s correlation analysis, the correlation between each species and the measured environmental factors was used to evaluate which species were affected by the environmental factors and the direction of their influence. Correlation heat maps were drawn at the phylum and genus levels to visually demonstrate correlations. In the figure, the abscissa is the measured related factors, and the ordinate is the flora at the classification level. Different colors represent the magnitude of the correlation coefficient, with darker blue indicating a stronger negative correlation and darker orange indicating a stronger positive correlation.

We used an absolute value of the correlation coefficient of >0.5 and *p* < 0.05 as the threshold for screening significant associations. The results showed that, at the phylum level, Firmicutes were positively correlated with ZO-1 and negatively correlated with LPS, IL-6, AST, and pSTAT3. Verrucomicrobia were negatively correlated with DAO. Proteobacteria were significantly positively correlated with COX-2 in the D1-T1 group. At the genus level, *Erysipelotrichrix* was significantly positively correlated with GM-CSF levels. *Lachnospira* was significantly negatively correlated with TNF-α. *Blautia* abundance was significantly negatively correlated with TLR-4. *Eubacterium* was significantly positively correlated with ZO-1 and negatively correlated with LPS. *Lactobacillus* was positively correlated with LPS, DAO, IL-6, TNF-α, AST, ALT, GM-CSF, NLRP3, pSTAT3, COX-2, and D-LA and negatively correlated with IL-10, TLR-4, ZO-1, occludin, and claudin-1. *Romboutsia* was positively correlated with ALT and GM-CSF levels and negatively correlated with occludin levels. *Christensenella* was significantly positively correlated with GM-CSF. *Clostridium* was significantly negatively correlated with LPS and pSTAT3 levels. *Ruminococcus* was positively correlated with ZO-1 and negatively correlated with LPS and pSTAT3 levels. *Oscillibacter* was significantly negatively correlated with LPS and TNF-α. *Akkermansia* was significantly negatively correlated with DAO. *Campylobacter* was positively correlated with GM-CSF. *Desulfovibrio* was significantly positively correlated with ZO-1 and negatively correlated with LPS, DAO, TNF-α, ALT, and pSTAT3. *Bilophilia* was significantly positively correlated with GM-CSF and D-LA levels. *Bifidobacterium* levels were significantly positively correlated with NLRP3, COX-2, and D-LA levels. *Rikenella* was significantly and positively correlated with MUC2 expression. *Prevotella* was significantly positively correlated with ALT and GM-CSF levels. *Alistipes* was significantly positively correlated with IL-10 and claudin-1 and was significantly negatively correlated with LPS, DAO, sIgA, IL-6, TNF-α, AST, NLRP3, pSTAT3, and COX-2. *Barnesiella* was significantly positively correlated with ZO-1 and claudin-1, and negatively correlated with LPS, DAO, IL-6, TNF-α, AST, GM-CSF, pSTAT3, COX-2, and D-LA (Figure 9). Species in the D7-T7 groups showed similar correlations to the measured environmental factors (Figure 10).

### 3.6. Dynamic Analysis

Dynamic comparison of the GM changes at T1d, T4d, and T7d in the experimental group showed that, with the extension of time, the observed Chao1 and ACE indices of the three groups all decreased at first and then increased and that the Shannon index gradually decreased. The species richness of the GM in the experimental group first decreased and then recovered slightly, whereas diversity decreased. There was no significant difference in alpha diversity between the three groups (*p* > 0.05). NMDS and PerMANOVA showed that there were significant differences in bacterial community composition between groups (*p* = 0.027), between T1d and T4d groups (*p* = 0.030), and between T1d and T7d groups (*p* = 0.035). However, the T4d and T7d groups were not significantly different (*p* = 0.467) (Figure 11). 

Venn diagram shows that there were 3557, 3283, and 3395 OUTs in the T1d, T4d, and T7d groups, respectively, indicating that the number of bacterial species in the samples after chemotherapy decreased first and then increased (Figure 12).

At the phylum level, the relative abundance of Firmicutes first decreased and then increased with time. The abundance was the lowest on post-operative day (POD) 4 and increased slightly on POD 7 but was still lower than that on POD 1. The relative abundances of Proteobacteria first increased and then decreased with time. The abundance reached a peak at POD 4 and decreased slightly at POD 7 but was still higher than that at POD 1. The relative abundance of Verrucomicrobia and Bacteroidetes increased first and then decreased, and reached the peak at POD 4, then decreased slightly at POD 7, but was still higher than that on POD 1. The relative abundance of Proteobacteria and Synergistetes gradually increased with time, but the magnitude gradually slowed down. At the genus level, the overall composition of the gut microbiota of New Zealand white rabbits showed a trend of the relative abundance of beneficial bacteria decreased first and then increased, while the relative abundance of harmful bacteria increased first and then decreased. The relative abundance of the main dominant genera was as follows: *Christinella*, *Ruminococcus*, and *Shigella* increased gradually, but the amplitude gradually slowed down. *Bacteroides* and *Parabacteroides* decreased gradually, but the range gradually slowed down. *Akkermansia* and *Clostridium* increased first and then decreased, but they were still higher in T7d group than in T1d group. *Enterococcus* and *Muribaculaceae* decreased first and then increased but were still lower in T7d group than in T1d group (Figure 13). 

## 4. Discussion

Changes in the GM, inflammatory factors, and intestinal barrier protein indices in New Zealand white rabbits were compared before and after injection of the chemotherapeutic drug, epirubicin, and a correlation analysis was performed. We also analyzed dynamic changes in the GM over time in the chemotherapy (experimental) group. The results showed that Firmicutes, Proteobacteria, and Bacteroidetes were the main phyla in the control and experimental groups, and that the GM diversity changed significantly after epirubicin treatment. The rabbit GM also exhibited dynamic changes at different time points following chemotherapy.

The abundance and diversity of the GM in the experimental group decreased and then increased with time after chemotherapy compared with that in the control group. The abundance of Firmicutes decreased, that of Proteobacteria increased, and that of beneficial bacteria, such as Muribaculaceae, *Enterococcus*, and *Ruminococcus*, decreased in the experimental group compared with that in the control group. The levels of harmful bacterial genera, such as *Bacteroides* and *Escherichia shigella*, also increased. In addition, with the passage of time after chemotherapy, *Ruminococcus* and *E. shigella* strains showed a gradual upward trend. Bacteroides, Parabacteroides, and other GM showed a gradual declining trend, whereas certain GM showed a trend of first aggravation followed by partial recovery. *Ruminococcus* is a widely studied strain that converts primary bile acids to secondary ones, and its abundance is significantly decreased in patients with intestinal and liver diseases [11,12]. *Bacteroides* participate in several important metabolic activities in the human colon, including carbohydrate fermentation and nitrogen utilization, and play an important role in maintaining GM homeostasis. *Bacteroides* are relatively abundant in patients with inflammatory bowel and liver diseases [13,14]. In this study, chemotherapy inflicted partial liver damage and intestinal inflammation. The relative abundance of *Ruminococcus* decreased, that of Bacteroides increased, and that of the other GM changed significantly in the experimental group, proving that transarterial chemotherapy drug administration changes the structural composition of the GM in rabbits.

To explore the effects of changes in the GM after transarterial chemotherapy and the possible mechanism of the gut–liver axis, we analyzed serological and colonic tissue markers in rabbits. Compared with the control group, the serum ALT and AST levels in the experimental group were significantly increased, which may be related to the epirubicin-damaged liver tissue [15]. In terms of intestinal permeability, DAO is released into the blood in response to intestinal mucosal ischemia, congestion, or injury [16]. D-LA is mainly produced and metabolized by GM (*Streptococcus* and *Enterococcus*). When the intestinal barrier is intact, D-LA cannot enter the body and has no notable effects on healthy hosts. However, with GM dysbiosis and intestinal leakage, the level of D-LA increases and can enter the blood via the intestinal barrier [17]. Therefore, DAO and D-LA are ideal indicators of the integrity and degree of damage to the intestinal mucosal mechanical barrier, which is a key link in the function of the intestinal mucosal barrier [16,17]. In addition, ZO-1, occludin, and claudin-1 expression levels directly reflect the intestinal mucosal barrier function [18]. In this study, the serum and colon tissue levels of DAO, D-LA, ZO-1, and occludin were significantly higher in the experimental group than those in the control group, indicating that epirubicin disrupted intestinal barrier function and increased intestinal permeability. While chemotherapy kills tumor cells, it also increases apoptosis in normal tissues; however, these liver injuries are usually resolved in a week [19,20]. These findings led us to ponder whether liver function could be improved and restored in the hepatointestinal circulation after chemotherapy drugs and how it influences the state and function of the intestine.

Dynamic comparison revealed that, with the metabolism of epirubicin in the host body, the liver function recovery, intestinal barrier function improvements, and AST, DAO, and D-LA levels gradually decreased with time after chemotherapy. AST levels were close to those in the control group, and DAO and D-LA levels were still significantly higher than those in the control group. These results indicate that the recovery of intestinal barrier function occurs later than that of the liver function after chemotherapy.

LPS and TLR-4 play important roles in tumor development. Furthermore, endotoxins are important mediators of the interaction between the liver and intestines [21]. Epirubicin entering the liver through the hepatic artery causes partial hepatocyte damage and increases intestinal permeability, which causes GM dysbiosis, which in turn induces intestinal endotoxemia and toxins are released into the liver, forming a vicious cycle [22]. A GM imbalance stimulates the liver to transport active mediators from the biliary tract to the intestine, such as sIgA, to control the excessive growth of pathogenic bacteria and maintain intestinal homeostasis [23]. Intestinal endotoxins, such as pathogen-associated molecular patterns, enter the liver through the portal vein system, activate TLR-4 in the liver, and produce inflammatory factors that participate in the development of various liver diseases [23]. COX-2 is a key factor that triggers inflammatory responses. The LPS ligand of TLR-4 activates the STAT3 pathway, which activates the COX-2 signaling axis, causes local inflammation, and increases chemotherapy resistance, increasing the toxic side effects of clinical chemotherapy drugs, such as epirubicin, nausea, and diarrhea [24]. NF-kB is also an important downstream inflammatory signal of TLR-4, which can upregulate the expression of various inflammatory cytokines, including TNF-α, IL-6, and IL-8, and downregulate the expression of anti-inflammatory factor IL-10 [25].

Our study found that the levels of sIgA, LPS, pSTAT3, COX-2, TNF-α, IL-6, and IL-8 were significantly increased in the experimental group compared with those in the control group, while the IL-10 level was significantly decreased, and the TLR-4 and pNF-kB levels were not significantly changed. The changes in the abovementioned key signaling proteins and inflammatory factors were similar to those induced by LPS, suggesting that chemotherapy regulates LPS and TLR-4 expression via GM changes. With the gradual chemotherapy drug metabolism, these factors tend to return to normal levels. However, we can improve intestinal function by regulating the GM, improving the therapeutic effect of chemotherapy via these signaling pathways, and reducing gastrointestinal adverse reactions, such as nausea and diarrhea.

Our study has some limitations. Our sample size was small; therefore, future investigations should increase the sample size to improve data reliability. In addition, to study the effect of chemotherapeutic drugs on the host GM, we did not establish an HCC model according to the clinical state, which does not fully simulate the human disease state, making it difficult to fully represent normal GM changes. Owing to the limitations of conditions and techniques, we did not perform standard TACE in rabbits according to the clinical operation but adjusted the operation to open injection of chemotherapy drugs through the proper hepatic artery to study the effect of epirubicin on the GM. Surgical methods will inevitably have an impact on the GM and overall condition of rabbits. However, we controlled for a single variable between the control and experimental groups, namely epirubicin treatment, and maintained the same experimental procedure to reduce the influence of other factors.

## 5. Conclusions

Our findings suggest that there are some changes in the GM of rabbits after transarterial chemotherapy: the relative abundance of beneficial bacteria, such as *Muribaculaceae*, *Enterococcus*, *Ruminococcus*, and *Clostridia*, decreased in the experimental group compared with those in the control group. However, the relative abundance of harmful bacteria, such as *Bacteroides* and *Escherichia* (*Shigella*), was higher in the experimental group than that in the control group. This may be related to various factors, such as liver metabolism, intestinal barrier function, and hepatointestinal circulation. These findings agree with studies on signaling molecules downstream of LPS that have shown that chemotherapy indirectly affects its efficacy via the liver–gut axis. We provide novel insights regarding the clinical treatment of HCC. Finally, our results suggest that chemotherapy treatment efficacy and prognosis can be improved by using a combination of TLR-4 inhibitors, targeted GM regulation, or protection of the intestinal mucosa with chemotherapy.

## Figures and Tables

**Figure 1 biology-13-00230-f001:**
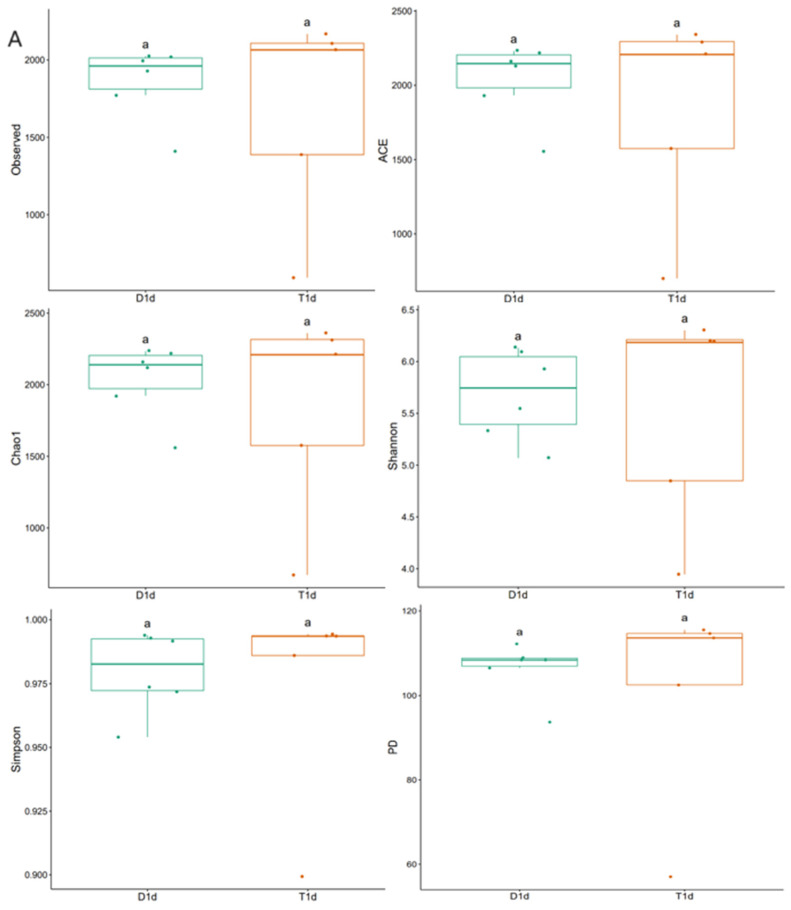
Comparison of alpha diversity indices between control and experimental groups based on observed, Chao1, ACE, Shannon, Simpson, and PD indicators. (**A**) Alpha diversity index comparison between D1d and T1d. (**B**) Alpha diversity index comparison between D7d and T7d. aa, no significant difference in the index between the two groups; ab, significant difference in the index between two groups (*p* < 0.05). D1d, control on post-operative day 1; T1d, chemotherapy on post-operative day 1; D7d, control on post-operative day 7; T7d, chemotherapy on post-operative day 7.

**Figure 2 biology-13-00230-f002:**
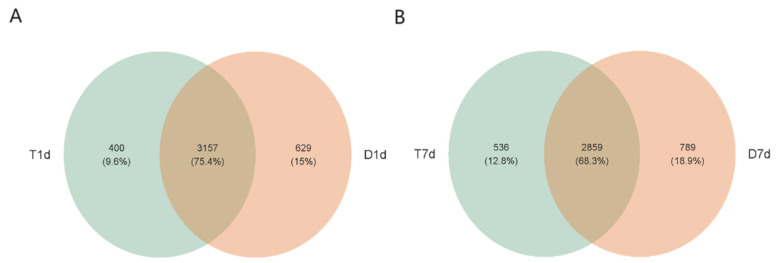
Venn plot of the control versus experimental groups at the OUTs level. (**A**) Venn diagram of D1d (orange) and T1d (green). (**B**) Venn diagram of D7d (orange) versus T7d (green). The overlap between two circles represents the common OUT between groups, the non-overlap represents the unique OUT in each group, and the number within each circle represents the number of unique OUTs in the circle.

**Figure 3 biology-13-00230-f003:**
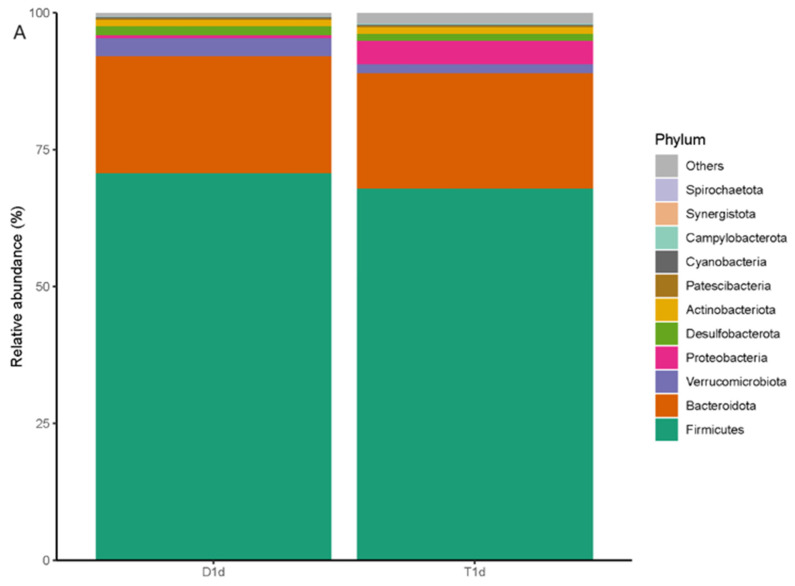
Bar chart comparisons of the composition and relative abundance of the microbiota at the phylum level between (**A**) D1d and T1d and (**B**) D7d and T7d. The colors represent different bacterial groups, and the proportion represents the relative abundance of each bacterial group. The comparison of the two groups of bar charts can directly compare the changes in the relative abundance of each bacterial group.

**Figure 4 biology-13-00230-f004:**
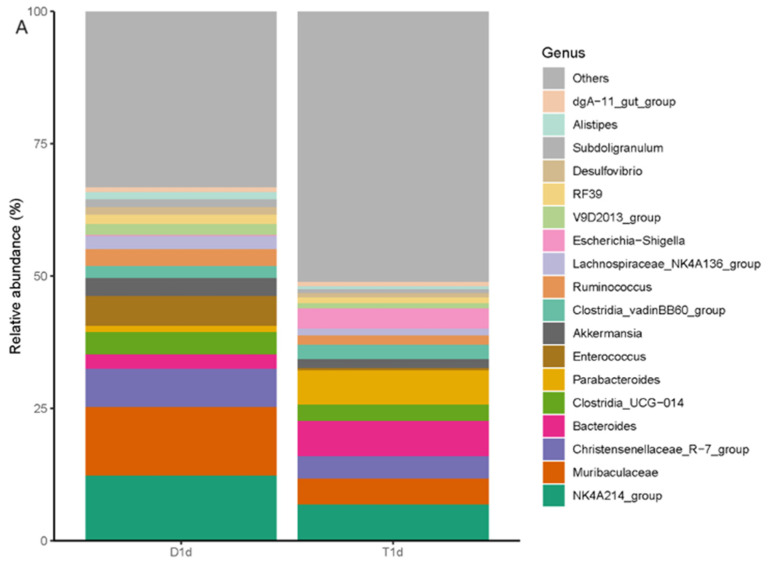
Bar chart comparison of the composition and relative abundance of the microbiota at the genus level between (**A**) D1d and T1d and (**B**) D7d and T7d. The colors represent different bacterial groups, and the proportion represents the relative abundance of each bacterial group. The comparison of the two groups of bar charts can directly compare the changes in the relative abundance of each bacterial group.

**Figure 5 biology-13-00230-f005:**
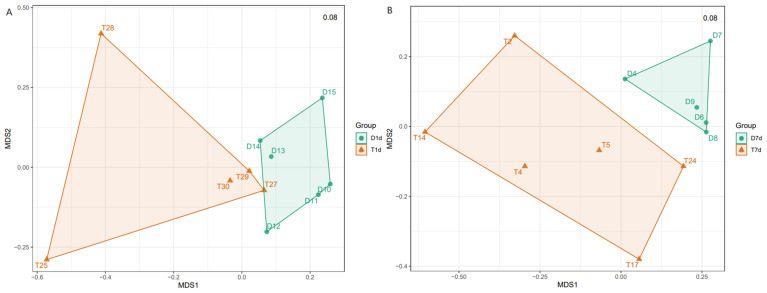
Beta diversity comparison between (**A**) D1d and T1d and (**B**) D7d and T7d. Each point in the figure represents a sample, and points of different colors belong to different groups. A significant difference in bacterial community composition can be observed between the D1d and T1d groups (*p* = 0.007) and D7d and T7d groups (*p* = 0.008).

**Figure 6 biology-13-00230-f006:**
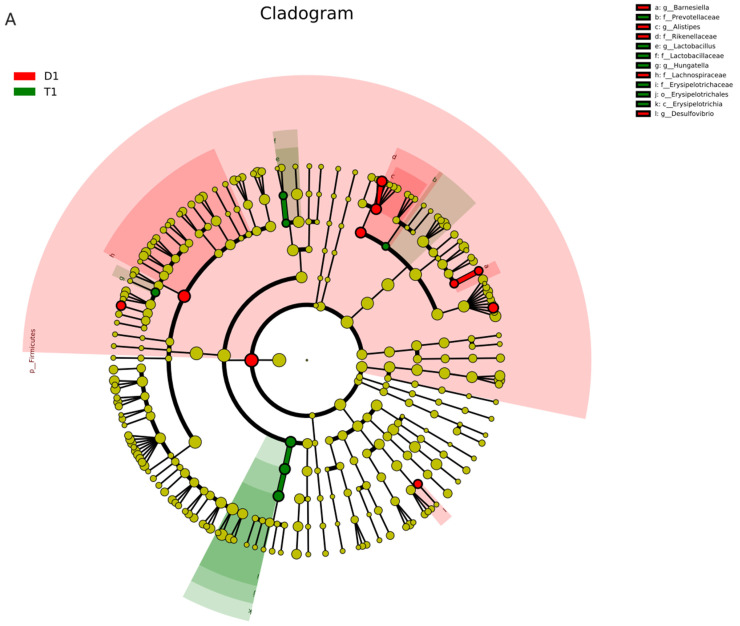
Linear discriminant analysis effect size analysis was used to screen out the significantly different species between groups, and the main bacteria in different taxonomic levels, such as phylum, class, order, family, and genus, are displayed on the species taxonomic branch map. Red, control group; green, experimental group. The taxonomic branch map of (**A**) D1d and T1d and (**B**) D7d and T7d.

**Figure 7 biology-13-00230-f007:**
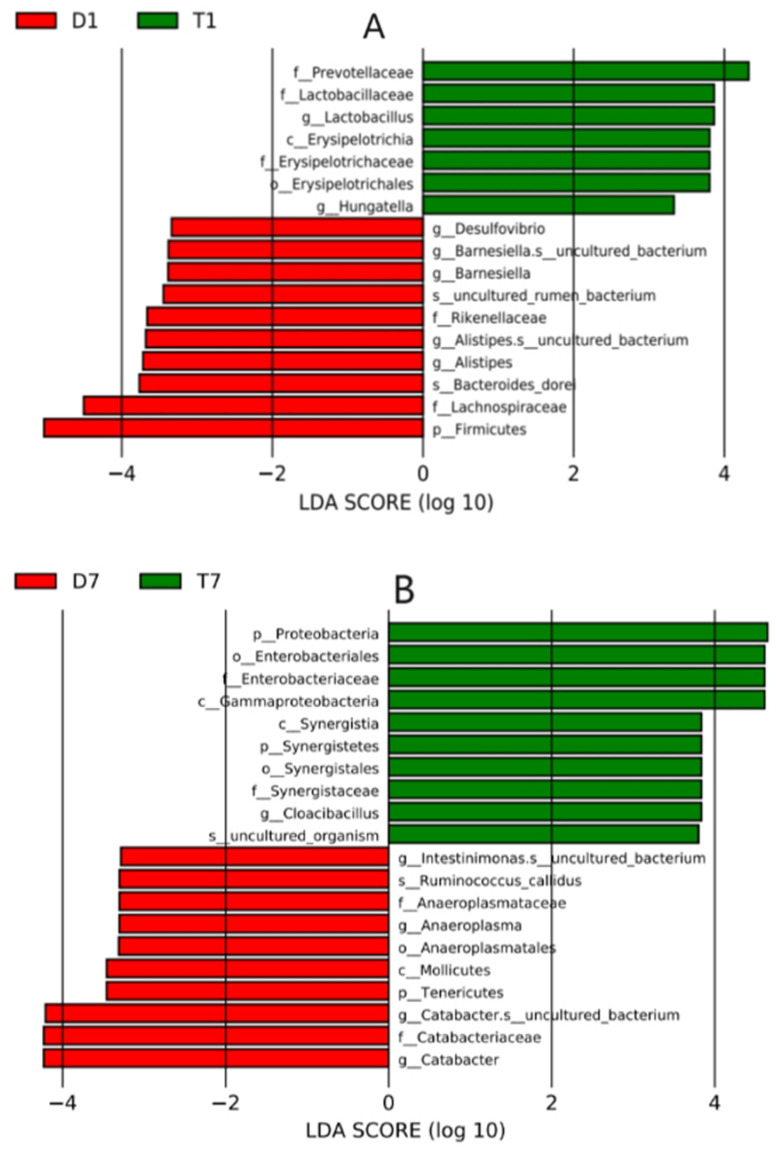
Linear discriminant analysis (LDA) was used to assess the maximum difference at different classification levels, as LDA score (log10) > 2 and *p* < 0.05 was used as the differential screening threshold. Red, control group; green, experimental group. LDA score of (**A**) D1d and T1d and (**B**) D7d and T7d.

**Figure 8 biology-13-00230-f008:**
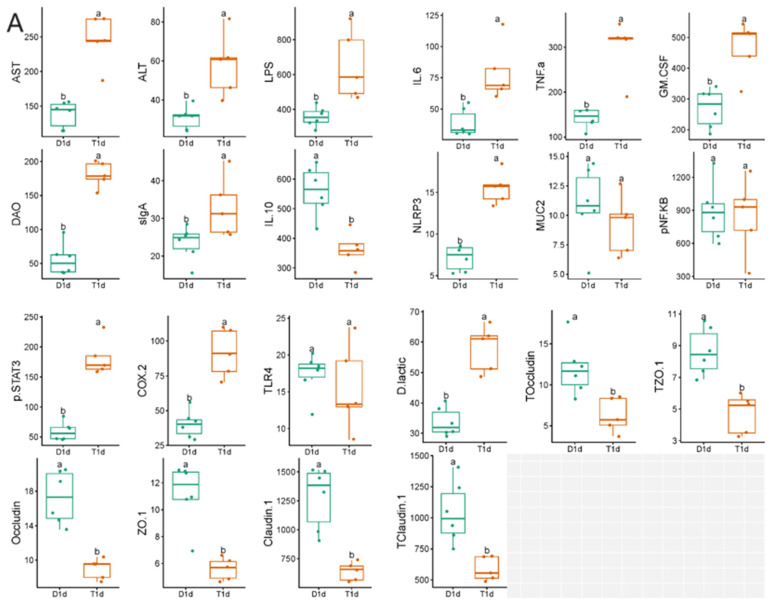
Comparison of the related indexes of rabbit serum and colon tissue measured using enzyme-linked immunosorbent assay (ELISA) between the control and experimental groups. aa, the index of the two groups with no significant difference; ab, the index of the two groups with significant difference (*p* < 0.05). The difference of each index between (**A**) D1d and T1d and (**B**) D7d and T7d.

**Figure 9 biology-13-00230-f009:**
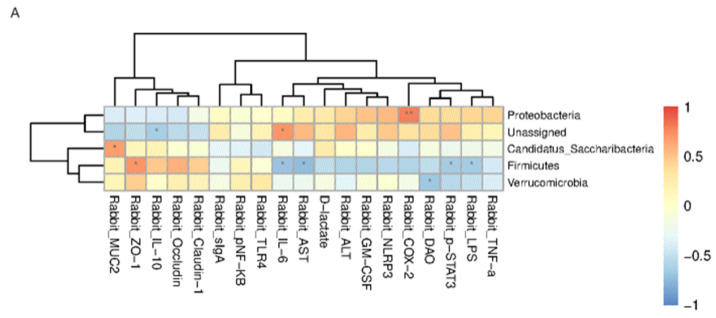
Spearman correlation analysis based on the relationship between each species and the measured environmental factors. Color intensity indicates the degree of correlation between the microbiota and environmental factors. The darker the color, the stronger the correlation; the lighter the color, the weaker the correlation. Orange represents positive correlation and blue represents negative correlation. * *p* < 0.05; ** *p* < 0.01. Spearman correlation analysis between (**A**) D1d and T1d at the gate level and (**B**) D1d and T1d at the genus level.

**Figure 10 biology-13-00230-f010:**
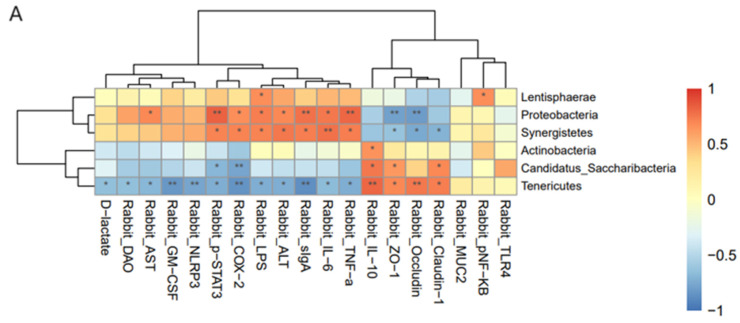
Spearman correlation analysis based on the relationship between each species and the measured environmental factors. Color intensity indicates the degree of correlation between the microbiota and environmental factors. The darker the color, the stronger the correlation. The lighter the color, the weaker the correlation. Orange represents positive correlation and blue represents negative correlation. * *p* < 0.05; ** *p* < 0.01. Spearman correlation analysis between (**A**) D7d and T7d at the phylum level and (**B**) D7d and T7d at the genus level.

**Figure 11 biology-13-00230-f011:**
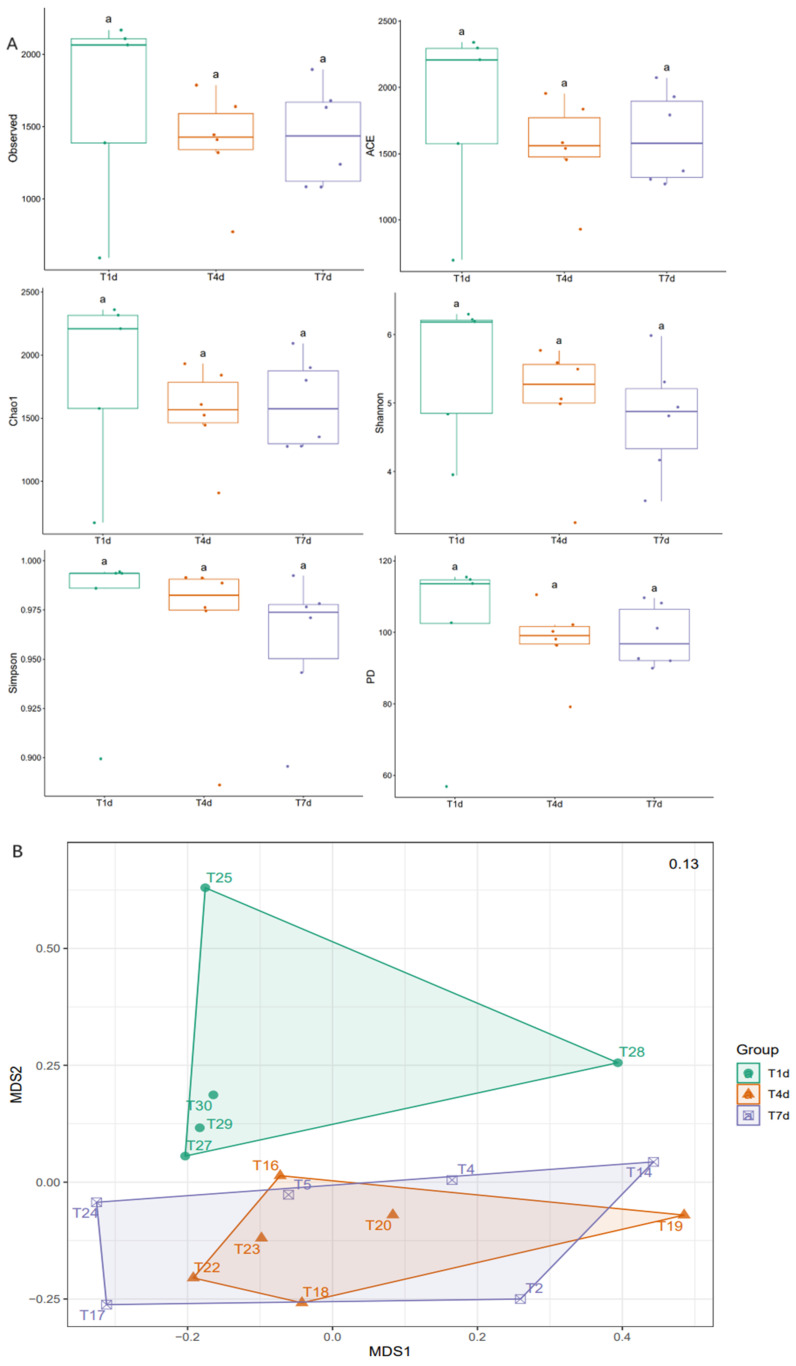
(**A**) Alpha diversity indices between the chemotherapy groups on observed, Chao1, ACE, Shannon, Simpson, and PD indicators. aaa, no significant difference in the index of the three groups, *p* > 0.05. (**B**) Beta diversity comparison between the chemotherapy groups. Each point in the figure represents a sample, and points of different colors belong to different groups. The bacterial community composition was significantly different between the T1d and T4d groups (*p* = 0.030), T1d and T7d groups (*p* = 0.035), and T4d and T7d groups (*p* = 0.467). T4d, chemotherapy group on post-operative day 4.

**Figure 12 biology-13-00230-f012:**
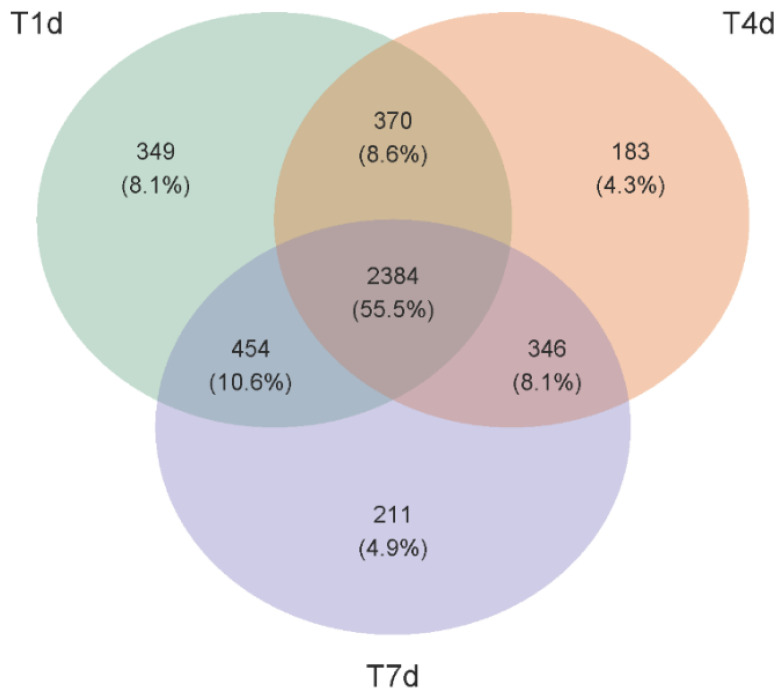
Venn plot of the comparison between the three chemotherapy groups at the OUTs level. Different circles represent the corresponding groups, the overlap between two circles represents the OUTs common between groups, non-overlap represents the OUTs unique to each group, and number within each circle represents the number of unique OUTs in the circle.

**Figure 13 biology-13-00230-f013:**
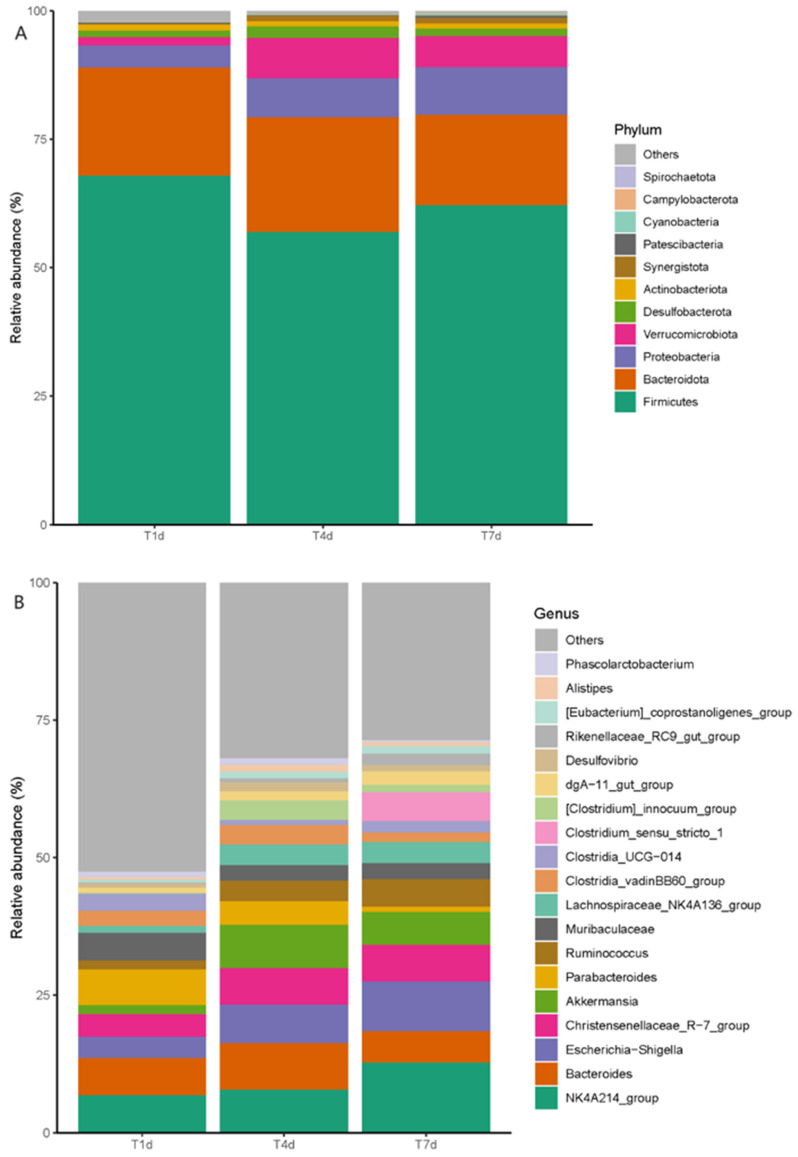
Bar chart comparison of microbiota composition and relative abundance at (**A**) phylum and (**B**) genus level for T1d, T4d, and T7d. The colors represent different bacterial groups, and the proportion represents the relative abundance of each bacterial group. The comparison of the two groups of bar charts can directly compare the changes in the relative abundance of each bacterial group.

## Data Availability

The data supporting the results of this study are included in this article. If any other data are needed, please contact the corresponding author.

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
