# Peer review of "Effect of Transarterial Chemotherapy on the Structure and Function of Gut Microbiota in New Zealand White Rabbits"

_biology, 2024, doi:10.3390/biology13040230_

Round 1

Reviewer 1 Report

Comments and Suggestions for Authors

Dear Authors,

First of all, congratulations on your research article!!!

However, I have identified some minor issues that I believe would benefit from your attention:

Title: I suggest considering including the term "white rabbit" in the title.

Introduction: It is satisfactory.

Materials and Methods:

MM1. Please specify the sex of the rabbits.

MM2. It would be beneficial to include a figure illustrating the experimental design, either within the main body or as Supplementary Material. Although well described, a visual aid would enhance comprehension for readers.

MM3. Please provide details on the primers used for 16S rRNA sequencing, the amplified region of the 16S gene, the bacterial DNA extraction method, and the sequencing technologies employed. Additionally, indicate the median and range of reads per treatment group (n=6).

MM4. Clarify the bioinformatic analysis conducted post-sequencing to generate the OTU table. Did you work with OTUs or ASVs? Specify the bioinformatic pipeline employed, whether Qiime2, Dada2, or another. Please provide clarification.

MM5. It appears from certain graphics in the Results section that R software and various R package libraries were utilized. Please specify the R packages employed in your study.

Results:

R1. Which statistical test was utilized to compare groups in terms of alpha diversity? Was rarefaction performed prior to calculating alpha diversity? Please state the statistical test in the Materials and Methods section for clarity. Thank you.

Discussion and Conclusions:

The Discussion and Conclusions sections are satisfactory. However, I recommend explicitly mentioning findings related to genus changes between treatments, in addition to the analyzed biomarkers, to highlight key discoveries to readers. It would be beneficial to include this information in the abstract as well.

Thank you.

Reviewer 2 Report

Comments and Suggestions for Authors

This study explores how chemotherapy during TACE treatment for HCC affects the gut microbiota (GM) in rabbits. Results show a shift from beneficial to harmful bacteria after treatment, peaking on the fourth day post-surgery and partially recovering by the seventh day. These findings suggest potential ways to manage HCC and reduce gastrointestinal side effects by targeting the GM alongside TACE.

Main Flaws:

1. While the authors utilized 16S rRNA sequencing to assess gut microbiota composition, essential methodological details such as the sequencing region, platform, and analysis pipelines were omitted. Additionally, they inaccurately referred to Operational Taxonomic Units (OTUs) as “species”, overlooking the fact that one OTU can represent multiple closely related species.

2. The lack of crucial experimental design details led to confusion for readers attempting to discern the comparisons being made. For example, at line 150 and line 294.

3. Inadequate data presentation, particularly the reliance on single mean values without statistical significance, hinders the ability to draw meaningful conclusions. Proper statistical analysis and the presentation of individual data points are necessary.

4. The terms "harmful" and "beneficial" bacteria were misused, as not all bacteria within Firmicutes or Proteobacteria are inherently good or bad.

Minor Points:

1. Clarification is needed regarding the "biological mechanism" and "theoretical basis" mentioned in line 38.

2. The term "quantity" in line 68 should be removed, as relative abundance comparisons were made, not quantifications.

3. Figure 3 legend should specify consistent color and order for easy comparison.

4. Uncertainty arises regarding the comparison of IL-10, ZO-1, occludin, and claudin-1 levels in line 233-234 (mRNA or protein).

5. Line 246-248 could better convey the authors' use of Spearman’s correlation analysis.

6. No statistically significant changes were observed in Figure 11A, contradicting the claim made in line 297.

7. "statistical analysis of " in line 312 should be replaced with "shows" for accuracy.

Comments on the Quality of English Language

The English overall is good, but it could benefit from refinement by a native speaker.
